# Associations Between Dietary Amino Acid Intake and Elevated High-Sensitivity C-Reactive Protein in Children: Insights from a Cross-Sectional Machine Learning Study

**DOI:** 10.3390/nu17132235

**Published:** 2025-07-05

**Authors:** Lianlong Yu, Xiaodong Zheng, Jilan Li, Changqing Liu, Yiya Liu, Meina Tian, Qianrang Zhu, Zhenchuang Tang, Maoyu Wu

**Affiliations:** 1Jinan Fruit Research Institute, All-China Federation of Supply and Marketing Co-Operatives, Jinan 250014, China; lianlong00a@163.com (L.Y.); zhxd1106@163.com (X.Z.); 15098844066@163.com (J.L.); 2Shandong Center for Disease Control and Prevention, Jinan 250014, China; 3Hebei Center for Disease Control and Prevention, Shijiazhuang 050021, China; lcq93@126.com (C.L.); tianmei78715@163.com (M.T.); 4Guizhou Center for Disease Control and Prevention, Guiyang 550004, China; liuyiya163@163.com; 5Jiangsu Provincial Center for Disease Control and Prevention, Nanjing 210028, China; zhuqianrang@hotmail.com; 6Institute of Food and Nutrition Development, Ministry of Agriculture and Rural Affairs, Beijing 100081, China

**Keywords:** dietary amino acids, hs-CRP, children, machine learning, cross-sectional study

## Abstract

**Background** High-sensitivity C-reactive protein (hs-CRP) is a protein that indicates inflammation and the risk of cardiovascular diseases. The intake of dietary amino acids can influence immune and inflammatory reactions. However, studies on the relationship between dietary amino acids and hs-CRP, especially in children, remain scarce. **Methods** This cross-sectional study analyzed data from the Nutrition and China Children and Lactating Women Nutrition and Health Survey (2016–2019), focusing on 3514 children (724 with elevated hs-CRP ≥ 3 mg/L and 2790 with normal levels). Dietary information was gathered via a food frequency questionnaire, and hs-CRP levels were obtained from blood samples. Boruta algorithm and propensity scores were used to select and match dietary factors and sample sizes. Machine learning (ML) algorithms and logistic regression models assessed the link between amino acid intake and elevated hs-CRP risk, adjusting for age, sex, BMI, and lifestyle factors. **Results** The odds ratios (ORs) for elevated hs-CRP were significant for several amino acids, including Ile, Leu, Lys, Ser, Cys, Tyr, His, Pro, SAA, and AAA, with values ranging from 1.10 to 2.07. The LightGBM algorithm was the most effective in predicting elevated hs-CRP risk, achieving an AUC of 0.927. Tyrosine, methionine, cysteine, and proline were identified as important features by SHAP analysis and logistic regression. The intake of Ser, Cys, Tyr, and Pro showed a linear increase in the risk of elevated hs-CRP, especially in individuals with low protein intake and normal weight (*p* < 0.1). **Conclusions** Intake of amino acids like Ser, Cys, Tyr, and Pro significantly impacts hs-CRP levels in children, indicating that regulating these could help prevent inflammation-related diseases. This study supports future dietary and health management strategies. This is first large-scale ML study linking amino acids to pediatric inflammation in China. The main limitations are the cross-section design and the use of self-reported dietary data.

## 1. Introduction

High-sensitivity C-reactive protein (hs-CRP) is a serum acute phase protein commonly utilized to evaluate the risk of inflammation and cardiovascular conditions [1,2]. Recent research indicates that hs-CRP levels are strongly linked to the development of several chronic conditions, such as obesity, diabetes, and cardiovascular disease [3,4,5]. In children and adolescents, elevated hs-CRP is also considered a sign of health risk [6,7], which may be closely related to multiple factors, such as nutritional status, lifestyle, and genetic factors [8,9].

Dietary amino acids, crucial for protein metabolism, have garnered increasing research interest lately due to their potential role in modulating immune and inflammatory responses [10,11]. In particular, some specific amino acids, such as valine, leucine, and lysine, have been shown to be associated with changes in inflammatory indicators in many studies [12,13]. Immune stress is closely related to the degradation pathway of leucine, isoleucine, and valine in the liver [12]. Despite some studies indicating a link between dietary protein intake and hs-CRP levels [14], research on the connection between dietary amino acids and hs-CRP, particularly in children, remains limited.

The objective of this research is to investigate how children’s amino acid consumption in their diet relates to the likelihood of increased hs-CRP levels, utilizing data from China Children and Lactating Women Nutrition and Health Survey. The study used a multistage cluster random sampling method to select data from children in different geographical regions, and through strict control of factors such as diet and lifestyle, it strives to reveal the possible relationship between dietary amino acids and inflammatory response. The results of the study will help provide a basis for dietary intervention in children and provide scientific support for the formulation of relevant public health policies.

## 2. Materials and Methods

### 2.1. Study Design and Participants

This study collected data from 275 surveillance sites of the China Children and Lactating Women Nutrition and Health Survey (CCLWNHS) (2016–2019) using multistage cluster random sampling, facilitated by the National Institute of Nutrition and Health under the Chinese CDC [15]. China’s 31 provinces were categorized into four groups: large cities, small and medium-sized cities, general rural counties, and poor rural counties, resulting in 124 strata. After data adjustments, 109 strata were finalized. Based on population distribution, 150 surveillance sites were chosen: 26 large cities, 45 small and medium-sized cities, 46 general rural counties, and 33 poor rural counties. Participants underwent interviews and physical exams and provided biological samples. Data from Shandong, Jiangsu, Hebei, and Guizhou were obtained from the CDC. Database searches occurred between October and December 2024, screening eligible participants. These provinces represent diverse dietary, socioeconomic, and lifestyle factors across China’s regions. Selecting these provinces ensures diverse exposure variables, representing children’s dietary habits across various Chinese regions. The survey employed consistent equipment and methods for data collection.

All study participants and their guardians gave written informed consent to take part. This study protocol was approved by the Ethics Review Committee of the NINH of the Chinese Center for Disease Control and Prevention (No. 201614). During the survey, a food frequency questionnaire (FFQ) with tested reliability and validity was used to collect the participants’ main dietary data for the past month [15]. The questionnaire was designed by a team of experts from the NINH and the Chinese Center for Disease Control and Prevention. It is a standardized questionnaire that has been validated and widely used in large-scale dietary surveys of Chinese children.

The project team contracted dietitians to collect information. The survey was conducted by trained statisticians through one-on-one, face-to-face interviews. The FFQ survey involved 12 food categories and 59 subcategories, including the frequency and amount of staple foods, beans, vegetables, fungi and algae, fruits, dairy products, meat, aquatic products, eggs, beverages, nutrient supplements and other foods, covering all major dietary categories of the Chinese population. In addition, the researchers weighed the cooking oils and condiments used in the participants’ homes or school cafeterias for three days in order to track the intake of key substances such as oil, salt, and monosodium glutamate.

The investigators were equipped with tablet computers, paper questionnaires, and dietary atlases, and received structured training on the questionnaire. Before the survey began, the investigators explained the purpose of the survey to the participants and asked about the consumption, frequency, and portion size of each food listed in the FFQ, covering the diet in the past month. For younger participants, the survey was completed with the assistance of their guardians. The intake of nutrients and amino acids was estimated according to the Standard Version of Chinese Food Composition Table (6th Edition) [16].

The main focus of this study was the occurrence of increased elevated hs-CRP. Elevated hs-CRP is defined as a high-sensitivity C-reactive protein (hs-CRP) concentration of ≥3 mg/L [17,18]. Participants provided fasting venous blood samples in the morning, and hs-CRP levels were assessed through an immunoturbidimetric assay. This method is based on the principle of particle-enhanced immunoturbidimetric assay, using dual liquid reagents and six-point calibration. The calibration standard is the traceable CRM470, and CRP quality control materials with normal and pathological values are provided. The detection range is 0.1–20 mg/L, the extended detection range is 0.1–300 mg/L, and the analytical sensitivity (lower limit of detection) is 0.03 mg/L. G & G TC-200K (Golden Human Nest (Beijing) Technology Co., Ltd., Beijing, China) was used for weight scale and Omron HBP-1300 was used for blood pressure monitoring. Food weighing was carried out using CAMRY, EK5350 (Golden Nest (Beijing) Technology Co., Ltd., Beijing, China). In addition, the study collected basic information on the participants, including their age, gender, and nationality, through a standardized questionnaire. Anthropometric measurements and fasting venous blood sample collection were performed using uniform methods and equipment. Based on previous studies, we considered a variety of confounders that may affect dietary intake or serum hs-CRP levels, including sex, age, body mass index (BMI), time spent outdoors, smoking or secondhand smoke exposure, alcohol consumption, dietary energy, dietary protein, dietary fat, percentage of energy from protein and fat, and percentage of carbohydrate intake, etc. Data on smoking, secondhand smoke exposure, and alcohol consumption were self-reported by participants. BMI was assessed using standardized equipment, and participants were sorted into normal weight, overweight, or obese groups according to the sex and age criteria [19]. Due to the rigorous sampling design employed in this study, we assumed that the missing data were random. To handle the missing data, we applied multiple imputation with chained equations (MICE) and created 20 data sets to fully utilize its ability to manage continuous and categorical variables and accommodate complex interrelationships. We selected the MICE method because it assumes data are missing at random (MAR), which was confirmed by our analysis of the missing data pattern. The multiple imputation model included all variables to thoroughly handle missing data.

### 2.2. Procedures

This study was based on CCLWNHS data from 2016 to 2019, involving 23,301 children and lactating women. First, according to the exclusion criteria, the following subjects were excluded: individuals aged over 18 years or younger than 6 years (10,452 people), individuals with missing dietary data (640 people), individuals with acute diseases (such as fever, infection, etc., 239 people), individuals with chronic kidney disease, malignant tumors, inborn metabolic diseases and other chronic diseases (579 people), and individuals with missing Hs-CRP data (812 people). Finally, the remaining 10,579 people entered the analysis. Next, the basic characteristics of the population and dietary factors (including various amino acids) were feature-selected using the Boruta algorithm after 100 iterations. Considering the low proportion of smoking exposure and drinking groups, after excluding dietary factors, 1:4 propensity score matching (PSM) was performed on elevated hs-CRP patients based on age, sex, BMI, and outdoor activity time, and the caliper value was set to 0.25. Finally, 3514 people were selected as the analysis subjects of the study, including 2790 people in the normal Hs-CRP group and 724 people in the elevated Hs-CRP group. This study did not use underweight and obesity as exclusion criteria because BMI changes dramatically during children’s growth and development. In order to maintain the continuity of BMI samples in the population, we used PSM and stratified analysis to exclude the influence of this segment of the population and investigate their specificity. Sampling exclusion was performed to remove missing data or outliers that could distort the analysis, thereby ensuring a more homogeneous and valid sample. The choice of a propensity score matching caliper value of 0.25 was based on standard practice, as it minimized data loss while ensuring reliable matching, ensuring comparability between treatment and control groups.

For the normal hs-CRP group (*n* = 2790) and elevated Hs-CRP group (*n* = 724), on the one hand, we applied four machine learning algorithms, extreme gradient boosting (XGBoost), light gradient boosting machine (LightGBM), naive Bayes (NB) and neural networks (NNs), and used the ROC–AUC curve to evaluate the best performance of each model, and at the same time, the SHAP algorithm was used to screen out more important variables. On the other hand, Logistic regression was used to analyze the binary dependent variables to achieve multi-model screening. Subsequently, the amino acid types with greater contributions were comprehensively screened, and finally verified by sensitivity analysis methods such as subgroup analysis, interaction analysis, and nonlinear analysis. Our data were matched by PSM to avoid unbalanced data. Therefore, the ROC curve is more suitable for this study than precision and recall curves.

### 2.3. Operational Algorithm Model

Propensity Score Matching (PSM) is a statistical technique employed to manage confounding variables in observational research and assess causal impacts. Its core principle is to simulate the effect of random grouping by calculating the probability (i.e., propensity score) of an individual receiving a certain intervention under given covariates (such as age, medical history, etc.) [20,21].

The Boruta algorithm identifies the most important features by comparing the z-value of each feature with the “shadow features”. The specific method involves copying all the real features and randomly shuffling them. In each iteration, the z-value of each feature is obtained from the random forest model, and the z-value of the shadow feature is generated by randomly shuffling the real features. When the z-value of a real feature is greater than the maximum z-value of the shadow features in multiple independent experiments, the real feature is considered to be “important” [22,23].

The XGBoost algorithm is an effective gradient boosting tree algorithm and belongs to the ensemble learning method based on decision trees. The algorithm gradually trains weak learners and processes the residuals of each prediction. In addition, by introducing regularization terms in the loss function, XGBoost can create a strong learner that is not prone to overfitting [24,25]. The hyperparameters were objective(“binary:logistic”), eval_metric(“auc”), max_depth(2), Eta(0.1), Lambda(1), and alpha(0).

LightGBM is a well-known boosting learning machine that combines multiple weak classifiers into a powerful classifier. LightGBM is an improved version of the gradient boosting decision tree (GBDT), which approximates the negative gradient of the residual of the current decision tree by repeatedly fitting new decision trees [26,27]. The hyperparameters were the learning_rate (0.1), matric (“auc”), lambda_l1 (1), lambda_l2 (1).

Naive Bayes (NB) is a classification algorithm that uses probability, relying on Bayes’ theorem and assuming that features are conditionally independent. Its core principle is to calculate the posterior probability of each category under a given feature (P(Y|X)) and select the category with the highest probability as the prediction result. The independence assumption between features greatly simplifies the calculation of joint probability (P(X|Y) = ∏ P(xi|Y)). This method is efficient and applicable to high-dimensional sparse data [28,29]. The hyperparameters were fL(0), usekernel(TRUE and adjust).

Neural Networks (NN) is a machine learning model that simulates the structure and information transmission mechanism of biological neurons. Through multi-layer connected “neuron” nodes (input layer, hidden layer, output layer), neural networks extract and combine features layer by layer, thereby achieving modeling of complex nonlinear relationships [30,31]. The hyperparameters were size(10), maxit(100), and decay(0.01).

Restricted Cubic Spline (RCS) is a statistical method for fitting nonlinear relationships between continuous variables and outcomes. Its principle is to construct a smooth curve through piecewise cubic polynomials and impose linear constraints at both ends of the data range to avoid overfitting [32,33].

The primary purpose of using machine learning in our research design was to use shap to screen and identify important variables. Because we performed PSM matching in this process, it is no longer suitable to split into a training set and validation set. We identified amino acid categories with large important contributions through machine learning and then used logistic regression to determine OR.

### 2.4. Statistical Analysis

In this research, binary variables were analyzed using either Fisher’s exact test or the chi-square test, while continuous variables were assessed using the Student’s *t*-test. We used four machine learning (ML) algorithms, including extreme gradient boosting (XGBoost), light gradient boosting machine (LightGBM), naive Bayes (NB), and neural networks (NN), to predict elevated hs-CRP risk. The model’s performance was assessed using the receiver operating characteristic (ROC) curve and the area under the curve (AUC) as evaluation metrics. For the optimal machine learning model, we used the Shapley Additive Explanation (SHAP) algorithm to further analyze the feature contribution of the optimal model by calculating the Shapley value of each variable based on game theory. When building the machine learning model, we used sex, age, BMI, time spent outdoors, smoking or exposure to secondhand smoke, alcohol consumption, dietary energy, dietary protein, dietary fat, percentage of energy intake from protein, percentage of energy intake from fat, percentage of energy intake from carbohydrates, and isoleucine (Ile), leucine (Leu), lysine (Lys), serine (Ser), cysteine (Cys), tyrosine (Tyr), phenylalanine (Phe), threonine (Thr), glycine(Gly), valine (Val), arginine (Arg), histidine (His), alanine (Ala), aspartic acid (Asp), glutamic acid (Glu), methionine (Met), proline (Pro), tryptophan (Trp), sulfur-containing amino acid (SAA), aromatic amino acid (AAA), and 34 other parameters. In addition, we also used logistic regression analysis to exclude collinear variables through multicollinearity test, and tested the odds ratios (ORs) and 95% confidence intervals (95% CI) of 20 amino acids on elevated hs-CRP after adjusting various confounding factors. Combined with machine learning, we screened out the top 15 features in terms of feature importance and selected key amino acid categories for sensitivity analysis. During the analysis, stratified analysis and interaction analysis were used, and restricted cubic spline (RCS) was used for the nonlinear test. All statistical analyses were performed using R software (version 4.3.1), and the packages involved in the analysis included Boruta, MatchIt, stats, forestplot, car, xgboost, lightgbm, e1071, e1071, and pROC, etc. The analysis was performed using R (version 4.2.0).

## 3. Results

### 3.1. Characteristics of Participants

In this study, 10,579 children and adolescents were screened through the inclusion and exclusion criteria (Figure 1). The Boruta algorithm validation screening found that the effects of all dietary factors on children’s elevated hs-CRP were consistent with the selective characteristics (Figure 2). The final sample was further matched by a 1:4 PSM (caliper value 0.25) (Figure 3). A total of 3514 people were included, including 2790 people in the normal hs-CRP group, accounting for 79.40%, and 724 people in the elevated hs-CRP group, accounting for 20.60%. The intake of Ile, Leu, Lys, Ser, Tyr, Phe, Thr, Val, Asp, Met, SAA, AAA, and secondhand smoke exposure in several characteristic information between the two groups were statistically significant (*p* < 0.05), while the other factors were not statistically significant (*p* > 0.05), as shown in Table 1. The sample in this study was sampled according to the urban–rural ratio of the Chinese population, so it can better reflect the distribution of urban and rural population and has a high representability. This design helps to enhance the broad generalizability of the findings, despite differences in the proportion of participants between urban and rural areas.

### 3.2. Association Between Dietary Amino Acids Intake and Elevated hs-CRP

Table 2 shows that in the three types of logistic regression models that adjusted for different confounding factors, Ile, Leu, Lys, Ser, Cys, Tyr, His, Pro, SAA, and AAA can significantly increase the risk of elevated hs-CRP, and the results are relatively stable. Model 1 is a crude model without adjusting for confounding factors, and model 2 adjusted for age, sex, and BMI. After the multicollinearity test, all the confounding factors corrected by model 3 were determined. Model 3 was adjusted for age, sex, BMI, secondhand smoke exposure, alcohol consumption, time spent outdoors, and proportion of energy intake from protein. After sufficient adjustment for confounding factors, the ORs of Ile, Leu, Lys, Ser, Cys, Tyr, His, Pro, SAA, and AAA for Elevated hs-CRP were 1.24 (1.02~1.49), 1.10 (1.01~1.20), 1.15 (1.01~1.31), 1.17 (1.01~1.37), 2.07 (1.13~3.78), 1.26 (1.02~1.57), 1.52 (1.05~2.22), 1.17 (1.03~1.34), and 1.10 (1.01~1.20).

### 3.3. Machine Learning and Sensitivity Analysis

The Receiver Operating Characteristic (ROC) of the four machine learning (ML) algorithms evaluated the performance of LightGBM with the best AUC = 0.927 (0.918–0.938) (Figure 4). For the best-performing machine learning model, LightGBM, the Shapley additive explanation (SHAP) algorithm was employed to compute the Shapley value of each variable using game theory to provide further insights into the model. The SHAP method was used to interpret and assess machine learning models, highlighting the significance of each variable in the LightGBM model. The results showed that the top 15 amino acid intakes associated with the risk of elevated hs-CRP were Tyr, Met, Cys, Pro, Ala, Gly, Arg, and Ser (Figure 5).

Combining the results of LightGBM and logistic regression, Ser, Cys, Tyr, and Pro met the top 50% of the importance of machine learning features, and logistic regression proved that they had an increased risk of elevated hs-CRP (*p* < 0.05). A sensitivity analysis was performed on the relationship between Ser, Cys, Tyr, Pro and elevated hs-CRP. Through stratified analysis, it was found that in the low-dose protein intake group protein (%) < P50 (23%), the effect of Ser, Tyr, and Pro on elevated hs-CRP was relatively stable (*p* < 0.05). In the normal-weight group, the effect of Cys and Tyr on elevated hs-CRP was more stable (*p* < 0.05) (Figure 6). Age, sex, weight status, protein percentage and other confounding factors had no statistically significant interaction on the relationship between these amino acids and elevated hs-CRP (*p* > 0.05). The RCS results showed that the risk of elevated hs-CRP increased linearly with the increase of Ser, Cys, Tyr, and Pro intake (*p* for nonlinear > 0.05) (Figure 7).

## 4. Discussion

The main finding of this study is that the intake of multiple amino acids in children’s diet is significantly associated with elevated high-sensitivity C-reactive protein (hs-CRP), especially the intake of amino acids such as Ser, Cys, Tyr, and Pro, which may increase the risk of children developing elevated hs-CRP. These findings suggest that dietary amino acids may play an important role in children’s inflammatory response, thereby affecting the risk of chronic diseases.

Compared with previous studies, our study provided a wider population sample and regional data and adopted more sophisticated analytical methods (such as machine learning and propensity score matching) [34], providing new evidence for the relationship between dietary amino acids and hs-CRP. Many previous studies focused on the effects of specific amino acids on adult populations or were limited to children in a specific region. This study covered children from different regions of China and, through comprehensive analysis of a variety of potential confounding factors, reached some more stable and widely applicable conclusions.

The results of this study indicate that there is an association between the intake of certain amino acids in the diet and increased levels of hs-CRP in children, which may affect children’s health by regulating the inflammatory response of the immune system. Previous studies have investigated the association between dietary protein intake and hs-CRP levels [14]; however, they did not examine the specific role of individual amino acids within proteins. Our study refines this analysis by focusing on the specific contributions of amino acids, addressing this gap and providing a more precise understanding of their potential impact on hs-CRP levels. This finding is consistent with the results of some basic studies that have shown that amino acids, especially branched-chain amino acids (such as leucine and isoleucine), play an important role in regulating cellular immune responses and activating inflammatory pathways [35]. Leucine and isoleucine are not only involved in protein synthesis but may also affect the production of inflammatory factors by activating the mTOR signaling pathway [36]. In addition, this study found that the relationship between cysteine and increased hs-CRP was more significant, which is closely related to its role as a sulfide reductase in the body [37]. These results provide further evidence for the role of dietary amino acids in regulating children’s immune and inflammatory responses. The relationship was more stable in the normal-weight group, probably because overweight and obesity may cause metabolic disturbances that interfere with the association with the study variables.

For clinicians and public health decision makers, these findings suggest that improving the amino acid intake of children’s diets, especially controlling the excessive intake of certain amino acids, may be an effective strategy for preventing diseases related to inflammation. In particular, in groups with higher levels of inflammation, appropriate dietary interventions may help reduce the risk of related diseases [37]. However, these recommendations should be formulated based on individual circumstances, such as age, gender, nutritional status, etc., to maximize the effect of dietary interventions.

There are few studies on the role of amino acids in the inflammatory response in children. A study conducted on children in North America revealed that the increased demand for amino acids (AAs) due to environmental enteric dysfunction (EED) amplifies the changes in AA needs brought about by rapid growth or acute infections [38]. Similarly, a study on adults in India indicated that individuals who are chronically undernourished require up to 50% more essential amino acids, such as lysine, compared to those who are well-nourished [39]. These studies only illustrate the metabolic interactions of amino acids during inflammation.

## 5. Limitations

Although this study provides meaningful results, there are still some limitations and unanswered questions. First, although we used multiple statistical methods to control most of the confounding factors, due to the limitations of the study design, it is impossible to completely exclude all possible confounding effects. Some residual confounding effects (such as genetic factors and mental health) cannot be excluded due to the lack of information. Future studies could adopt more sophisticated longitudinal designs to observe the causal relationship between long-term intake of dietary amino acids and inflammatory response. Second, this study mainly relies on food frequency questionnaires (FFQs) to estimate dietary intake, which may have certain biases, especially when children and guardians report dietary information. Additionally, the FFQ does not capture dietary changes in real time. To improve the accuracy of the data, future studies can combine more accurate dietary records or biomarker detection. In addition, although we studied the intake of multiple amino acids, the interactions of amino acids or the joint effects of multiple dietary components have not been considered. Future studies could further explore the combined effects of these factors.

## 6. Conclusions

This study showed that there was a significant positive correlation between dietary amino acid intake and children’s hs-CRP levels, among which the more important amino acids included Ser, Cys, Tyr, and Pro. This finding provides a new perspective for understanding the relationship between children’s nutrition and inflammatory response, and provides theoretical support for future dietary intervention and health management.

## Figures and Tables

**Figure 1 nutrients-17-02235-f001:**
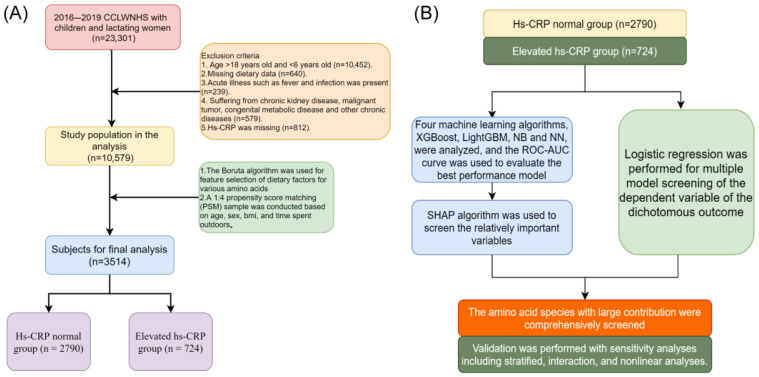
Study flow chart. (**A**) flow chart of patient inclusion and exclusion; (**B**) a research flowchart relying on machine learning.

**Figure 2 nutrients-17-02235-f002:**
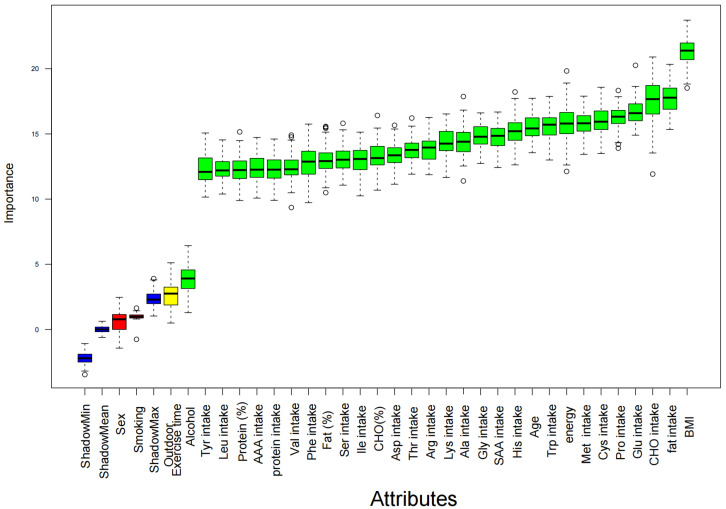
The outcome of Boruta for elevated hs-CRP. Green, yellow, and blue boxplots were identified as significant, tentative, and insignificant variables, respectively.

**Figure 3 nutrients-17-02235-f003:**
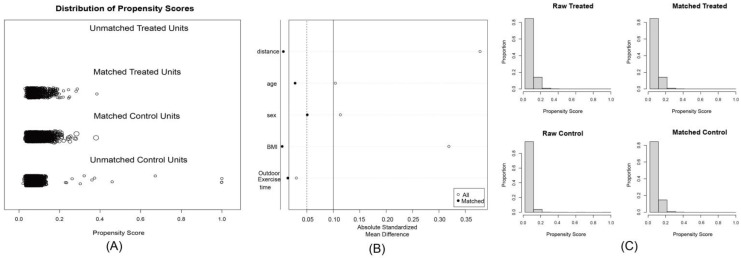
(**A**) Distribution results of patients after propensity score matching. (**B**) Love plot comparing absolute standardized mean differences in propensity scores and covariates between unmatched and matched samples. (**C**) histogram of patient distribution before and after propensity score matching.

**Figure 4 nutrients-17-02235-f004:**
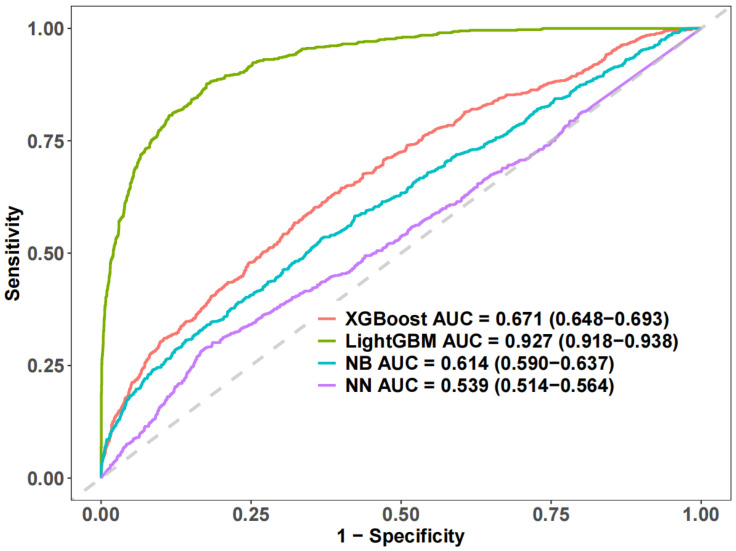
Receiver operating characteristic (ROC) curve of the machine learning model.

**Figure 5 nutrients-17-02235-f005:**
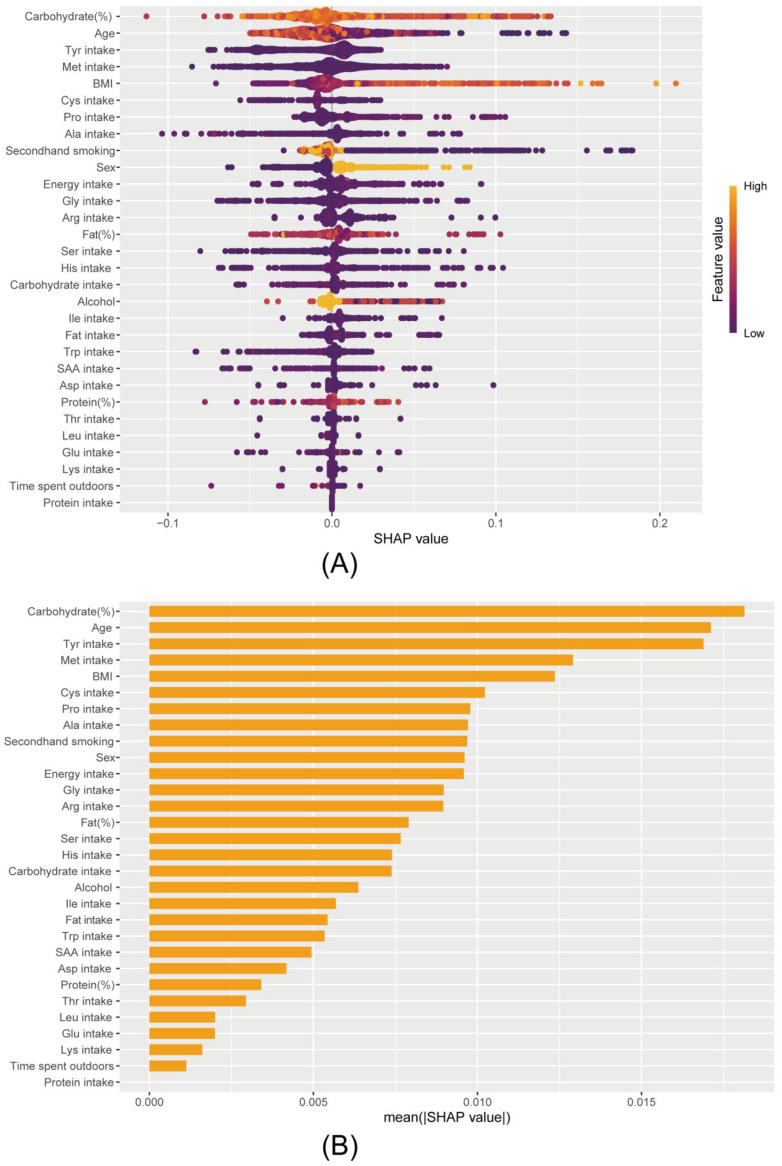
Interpretation and evaluation of machine learning models based on LightGBM. (**A**) SHAP analyses the dataset, which shows the 30 most important features and their impact on the model output. Each point represents a patient; purple represents the lowest range and yellow the range of the highest range features; (**B**) SHAP analysis feature importance ranking.

**Figure 6 nutrients-17-02235-f006:**
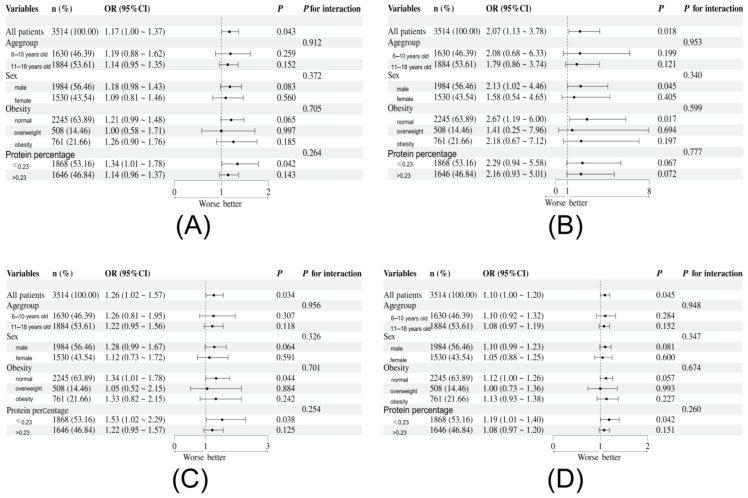
Stratified analysis forest plot of the association between multiple categories of dietary amino acids and Elevated hs-CRP risk. The interaction of each stratification factor on the above relationship was shown in the last column. Amino acid species: (**A**) Ser; (**B**) Cys; (**C**) Tyr; (**D**) Pro. OR, odds ratio, CI, confidence interval.

**Figure 7 nutrients-17-02235-f007:**
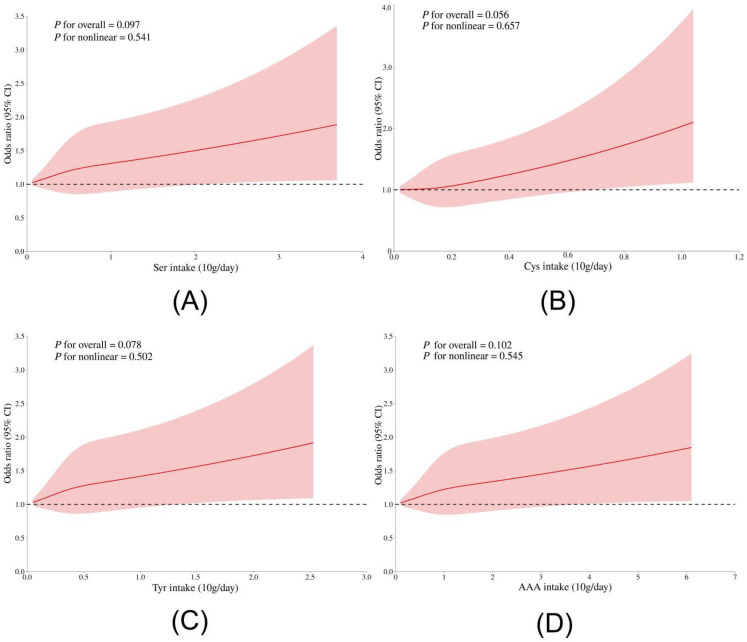
RCS analysis of different amino acid intake and the risk of elevated hs-CRP. The model was adjusted for age, sex, BMI, secondhand smoke exposure, alcohol consumption, time spent outdoors, and proportion of energy intake from protein. Amino acid species: (**A**) Ser; (**B**) Cys; (**C**) Tyr; (**D**) Pro. OR, odds ratio, CI, confidence interval.

**Table 1 nutrients-17-02235-t001:** Baseline Features of the Study population.

Variables	Total (*n =* 3514)	hs-CRP Normal Group (*n =* 2790)	Elevated hs-CRP Group (*n =* 724)	Statistic	*p* ^1,2^
Age (years)	11.10 ± 3.16	11.12 ± 3.14	11.01 ± 3.24	t = 0.79	0.43
Sex, *n* (%)				χ^2^ = 0.43	0.513
male	1984 (56.46)	1583 (56.74)	401 (55.39)		
female	1530 (43.54)	1207 (43.26)	323 (44.61)		
BMI (kg/m^2^)	20.26 ± 4.90	20.18 ± 4.81	20.58 ± 5.26	t = −1.84	0.066
Alcohol, *n* (%)				χ^2^ = 5.63	0.06
Yes, within 30 days	139 (3.96)	104 (3.73)	35 (4.83)		
Yes, before the last 30 days	228 (6.49)	170 (6.09)	58 (8.01)		
Never drank	3147 (89.56)	2516 (90.18)	631 (87.15)		
Secondhand smoke exposure, *n* (%)				χ^2^ = 9.88	0.042
everyday	334 (9.50)	253 (9.07)	81 (11.19)		
4–6 days per week	137 (3.90)	114 (4.09)	23 (3.18)		
1–3 days per week	402 (11.44)	305 (10.93)	97 (13.40)		
less than one day a week	511 (14.54)	421 (15.09)	90 (12.43)		
no	2130 (60.61)	1697 (60.82)	433 (59.81)		
Outdoor exercise time (hours)	1.96 ± 2.38	1.96 ± 2.42	1.95 ± 2.24	t = 0.10	0.923
Energy intake (kcal/day)	2381.28 ± 1964.52	2352.34 ± 1721.57	2492.77 ± 2702.51	t = −1.71	0.0866
Protein intake (g/day)	126.31 ± 121.35	124.26 ±95.01	134.22 ± 191.44	t = −1.97	0.00491
Carbohydrates intake (g/day)	361.82 ± 281.18	358.29 ± 243.55	375.40 ± 393.83	t = −1.46	0.145
fat intake (g/day)	47.64 ± 61.18	46.91 ± 60.19	50.48 ± 64.79	t = −1.40	0.162
Fat intake percentage	0.17± 0.08	0.16 ± 0.08	0.17 ± 0.08	t = −1.88	0.0601
Carbohydrates intake percentage	0.62 ± 0.09	0.62 ± 0.09	0.62 ± 0.10	t = 1.70	0.0888
Protein intake percentage	0.21 ± 0.05	0.21 ± 0.05	0.21± 0.05	t = −0.25	0.8034
Ile intake (10 g/day)	0.47 ± 0.41	0.46 ± 0.34	0.50 ± 0.61	t = −2.34	0.019
Leu intake (10 g/day)	0.95 ± 0.90	0.93 ± 0.71	1.01 ± 1.40	t = −2.22	0.027
Lys intake (10 g/day)	0.62 ± 0.61	0.61 ± 0.50	0.66 ± 0.91	t = −2.25	0.024
Ser intake (10 g/day)	0.56 ± 0.53	0.55 ± 0.41	0.60 ± 0.83	t = −2.23	0.026
Cys intake (10 g/day)	0.16 ± 0.13	0.16 ± 0.11	0.17 ± 0.17	t = −1.93	0.054
Tyr intake (10 g/day)	0.41 ± 0.37	0.41 ± 0.30	0.44 ± 0.57	t = −2.30	0.022
Phe intake (10 g/day)	0.57 ± 0.52	0.56 ± 0.42	0.61 ± 0.81	t = −2.11	0.035
Thr intake (10 g/day)	0.49 ± 0.49	0.48 ± 0.38	0.53 ± 0.79	t = −2.18	0.03
Gly intake (10 g/day)	0.60 ± 0.58	0.59 ± 0.45	0.63 ± 0.93	t = −1.65	0.098
Val intake (10 g/day)	0.63 ± 0.62	0.62 ± 0.48	0.67 ± 0.99	t = −2.15	0.031
Arg intake (10 g/day)	0.73 ± 0.67	0.72 ± 0.53	0.77 ± 1.04	t = −1.71	0.088
His intake (10 g/day)	0.23 ± 0.20	0.23 ± 0.18	0.25 ± 0.28	t = −1.82	0.07
Ala intake (10 g/day)	0.76 ± 0.84	0.75 ± 0.61	0.81 ± 1.41	t = −1.68	0.092
Asp intake (10 g/day)	1.02 ± 0.98	1.00 ± 0.78	1.08 ± 1.53	t = −2.01	**0.044**
Glu intake (10 g/day)	2.07 ± 1.64	2.04 ± 1.45	2.18 ± 2.22	t = −1.56	0.12
Met intake (10 g/day)	0.26 ± 0.28	0.26 ± 0.21	0.28 ± 0.45	t = −2.01	**0.045**
Pro intake (10 g/day)	0.68 ± 0.57	0.66 ± 0.48	0.72 ± 0.83	t = −1.89	0.059
Trp intake (10 g/day)	0.16 ± 0.17	0.16 ± 0.13	0.18 ± 0.27	t = −1.96	0.05
SAA intake (10 g/day)	0.42 ± 0.39	0.41 ± 0.31	0.45 ± 0.61	t = −2.23	**0.026**
AAA intake (10 g/day)	0.98 ± 0.89	0.97 ± 0.71	1.05 ± 1.38	t = −2.19	**0.028**

^1^. Two sample *t*-test (*p* < 0.05); ^2^. Pearson Chi-square (*p* < 0.05). *p* < 0.05 was considered statistically significant (in bold).

**Table 2 nutrients-17-02235-t002:** Association between various dietary amino acid intake and the risk of Elevated hs-CRP in Chinese children and adolescents.

	Model 1 OR (95% CI)	*p*	Model 2 OR (95% CI)	*p*	Model 3 OR (95% CI)	*p*
Ile	1.22 (1.02~1.47)	0.028	1.23 (1.02~1.48)	0.027	1.24 (1.02~1.49)	0.029
Leu	1.09 (1.01~1.19)	0.044	1.09 (1.01~1.19)	0.043	1.10 (1.01~1.20)	0.045
Lys	1.14 (1.01~1.29)	0.035	1.14 (1.01~1.29)	0.035	1.15 (1.01~1.31)	0.036
Ser	1.16 (1.01~1.34)	0.043	1.16 (1.01~1.35)	0.042	1.17 (1.01~1.37)	0.043
Cys	2.05 (1.15~3.67)	0.015	2.11 (1.17~3.82)	0.014	2.07 (1.13~3.78)	0.018
Tyr	1.25 (1.02~1.53)	0.034	1.25 (1.02~1.54)	0.032	1.26 (1.02~1.57)	0.034
Phe	1.15 (1.00~1.33)	0.052	1.16 (1.00~1.34)	0.052	1.16 (1.00~1.36)	0.054
Thr	1.17 (1.00~1.37)	0.052	1.17 (1.00~1.37)	0.051	1.18 (1.00~1.40)	0.053
Gly	1.10 (0.97~1.26)	0.130	1.10 (0.97~1.26)	0.132	1.11 (0.97~1.28)	0.133
Val	1.13 (1.00~1.28)	0.053	1.13 (1.00~1.28)	0.052	1.14 (1.00~1.30)	0.053
Arg	1.09 (0.98~1.22)	0.112	1.10 (0.98~1.22)	0.111	1.10 (0.98~1.24)	0.115
His	1.51 (1.06~2.16)	0.023	1.53 (1.06~2.19)	0.023	1.52 (1.05~2.22)	0.028
Ala	1.07 (0.98~1.17)	0.134	1.07 (0.98~1.17)	0.133	1.08 (0.98~1.19)	0.134
Asp	1.08 (1.00~1.16)	0.065	1.08 (1.00~1.16)	0.064	1.08 (1.00~1.17)	0.065
Glu	1.05 (1.00~1.09)	0.054	1.05 (1.00~1.10)	0.056	1.04 (1.00~1.09)	0.067
Met	1.29 (0.97~1.72)	0.076	1.29 (0.97~1.72)	0.075	1.32 (0.97~1.80)	0.075
Pro	1.17 (1.03~1.33)	0.015	1.18 (1.03~1.34)	0.015	1.17 (1.03~1.34)	0.018
Trp	1.51 (0.95~2.40)	0.081	1.52 (0.95~2.41)	0.079	1.55 (0.95~2.55)	0.081
GET	1.23 (1.01~1.49)	0.042	1.23 (1.01~1.50)	0.041	1.24 (1.01~1.53)	0.044
AAA	1.09 (1.01~1.19)	0.044	1.09 (1.01~1.19)	0.042	1.10 (1.01~1.20)	0.045

Logistic regression (*p* < 0.05). Abbreviations: CI, confidence interval; Model 1 unadjusted. Model 2 adjusted for age, sex, and BMI. Model 3 adjusted for age, sex, BMI, secondhand smoke exposure, alcohol consumption, time spent outdoors, and proportion of energy intake from protein. In these models, amino acids units were adjusted to 10 g/day. *p* < 0.05 was considered statistically significant.

## Data Availability

Since the copyright for the data in this study belongs to the Chinese Center for Disease Control, if there is a need to share data or code, please contact the first author and corresponding author, or visit https://www.chinacdc.cn/.

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
