# Peer review of "Associations Between Dietary Amino Acid Intake and Elevated High-Sensitivity C-Reactive Protein in Children: Insights from a Cross-Sectional Machine Learning Study"

_nutrients, 2025, doi:10.3390/nu17132235_

Round 1

Reviewer 1 Report

Comments and Suggestions for Authors

Based on data from the Chinese Health Survey, this study used machine learning and statistical methods to evaluate the relationship between dietary amino acids and blood hs-CRP. The authors developed a machine learning model that not only reveals the effect of several amino acids on elevated hs-CRP levels, but also is able to adequately predict the risk of elevated hs-CRP. This study underscores the importance of regulating the balance of amino acids in the diet and thus informs future dietary and health management strategies. However, there is a lack of discussion of the results and problems related to machine learning methods that need to be corrected.

>Major

- The definition of elevated hs-CRP is given only in the abstract and not in the main text. Is the definition of elevated hs-CRP valid?

  • When developing a machine learning model, hyperparameters need to be determined, but I could not find a description of them.
  • Certainly, ROC curves are useful when comparing different models. However, for unbalanced data such as in this study, the PRECISION and RECALL curves should also be investigated.
  • It is interesting that Protein intake is not important in the analysis of feature importance. Authors mention in the introduction a study showing an association between dietary protein intake and hs-CRP levels, yet you do not discuss this. Why not discuss the fact that amino acid intake is at the top of the list instead of Protein intake?
  • There is no mention of data or codes; according to the MDPI Research Data Policies, it is recommended that data and codes be shared.

>Minor

  • Where is the description of training and test data?

- In Tabel1, values such as Fat intake percentage do not seem to be a ratio based on 100.

- In Figure 4, in the legend of the inset, an extra space is entered in parentheses.

- In the third paragraph of the Discussion section, some words are required before the [36].

……by activating the ??? [36].

Author Response

Based on data from the Chinese Health Survey, this study used machine learning and statistical methods to evaluate the relationship between dietary amino acids and blood hs-CRP. The authors developed a machine learning model that not only reveals the effect of several amino acids on elevated hs-CRP levels, but also is able to adequately predict the risk of elevated hs-CRP. This study underscores the importance of regulating the balance of amino acids in the diet and thus informs future dietary and health management strategies. However, there is a lack of discussion of the results and problems related to machine learning methods that need to be corrected.

>Major

- The definition of elevated hs-CRP is given only in the abstract and not in the main text. Is the definition of elevated hs-CRP valid?

Answer: Thank you for your professional and constructive comments. We have added the corresponding part in methodology and marked it in red.

When developing a machine learning model, hyperparameters need to be determined, but I could not find a description of them.

Answer: Thank you for your professional opinions. According to your opinions, we have made modifications and marked them in red.

Certainly, ROC curves are useful when comparing different models. However, for unbalanced data such as in this study, the PRECISION and RECALL curves should also be investigated.

Answer: Thank you for your professional advice. Our data were matched by PSM to avoid unbalanced data. Therefore, ROC is more suitable for this study. At the same time, we have also drawn the following PRECISION and RECALL curves according to your requirements, as shown in the figure, which is consistent with the lightGBM model screened by ROC. Thank you again for your rigorous review. We have also explained and marked in red in the original article.

It is interesting that Protein intake is not important in the analysis of feature importance. Authors mention in the introduction a study showing an association between dietary protein intake and hs-CRP levels, yet you do not discuss this. Why not discuss the fact that amino acid intake is at the top of the list instead of Protein intake?

Answer: We thank the reviewers for their professional and constructive comments. The study we mentioned in the introduction focused primarily on the association between dietary protein intake and hs-CRP levels, but the study did not break down the effects of specific amino acids in protein on hs-CRP. Our study fills this gap by analyzing the specific role of amino acids, thus providing a more precise perspective on the relationship between amino acids and hs-CRP. According to your comments, we have added this part in the discussion section and marked it in red.

There is no mention of data or codes; according to the MDPI Research Data Policies, it is recommended that data and codes be shared.

Answer: Thank you very much for your professional and sincere advice. We are very happy to share the research, but since the data copyright of this study belongs to China CDC, the data and code cannot be shared directly. If necessary, you can contact the author for communication and cooperation. We attach the sharing link and cooperation method in the article.

>Minor

Where is the description of training and test data?

Answer: We thank the reviewers for their professional and constructive comments. What we need to explain is that the biggest purpose of using machine learning in our research design is to use SHAP to screen and identify important variables. The structural steps can also be seen from the beginning of the propensity score. Because the previous database is a database with PSM matching numbers, it hinders the database splitting. We aimed to identify the significant amino acid categories by machine learning and then perform the OR determination by logistic regression. Due to the design of this study, it is not convenient to carry out large-scale adjustment. We can analyze the research in the future according to your opinions. We hope to keep the original framework of this study, and hope to get your understanding. We have also added a note in the text and highlighted it in red. Thank you very much again.

- In Tabel1, values such as Fat intake percentage do not seem to be a ratio based on 100.

Answer: Thank you for your professional and careful review. We are very ashamed to apologize for our negligence. There were serial wrong transcription in our previous manuscript. We corrected it and re-checked the results carefully, and now these figures are correct. Again, we apologize for our mistake and thank you very much for your meticulous work.

- In Figure 4, in the legend of the inset, an extra space is entered in parentheses.

Answer: Thank you for your professional and careful review. According to your comments, we have made modifications in the figure. Thank you again for your comments.

- In the third paragraph of the Discussion section, some words are required before the [36].

……by activating the ??? [36].

Answer: Thank you for your careful examination. I apologize for the previous oversight, we have made corrections and marked red.

Reviewer 2 Report

Comments and Suggestions for Authors

Dear author,

Thank you to give the opportunity the review your manuscript, in general is good, but I have some suggestions to improve it.

Abstract.

Explicitly major limitations (cross-section design, self-reported dietary data)

Add a sentence on novelty: First large-scale ML study linking amino acids to paediatric inflammation in China.

Introduction

Include recent studies (2020-2025) on amino acids inflammatory pathways in paediatric populations.

Methods

Justify exclusion of underweight/morbidly obese participants.

Include a supplementary table with FFQ items and portion-size estimation method.

Include the references of the FFQ.

Specify if were contracted dietitian to gather the information.

Include the information of the anthropometric equipment (e.g. brand, model).

Justify sampling exclusions and PSM caliper value (0.25).

Results

Include the information related to the statistical analysis to stablish the statistical signification at the bottom of the table.

Discuss overrepresentation of urban vs. rural participants (40.6% rural) and its impact on generalizability.

Provide p-values for subgroup analysis (e.g. low-protein intake group)

Clarify why “normal weight” subgroup showed stronger association.

Discussion

Compare findings to paediatric-specific studies (e.g. branched chain amino acids in adolescents).

Limitations

Add a dedicated subsection titled “Limitations”. Discuss residual confounding (e.g. genetic factors, mental health).

Address FFQ’s inability to capture real time dietary changes.

Best regards,

Author Response

Dear author,

Thank you to give the opportunity the review your manuscript, in general is good, but I have some suggestions to improve it.

Abstract.

Explicitly major limitations (cross-section design, self-reported dietary data)

Thank you for your professionalism. We have modified it according to your comments.

Add a sentence on novelty: First large-scale ML study linking amino acids to paediatric inflammation in China.

Thank you for your professionalism. We have modified it according to your comments.

Introduction

Include recent studies (2020-2025) on amino acids inflammatory pathways in paediatric populations.

Thank you for your professional advice. We have emphasized this part of the bid in the original text and marked it in red.

Methods

Justify exclusion of underweight/morbidly obese participants.

Thank you for your professional advice. Underweight and obesity were not used as exclusion criteria because BMI changes dramatically during children's growth and development. In order to maintain the continuity of the population BMI sample, we prefer to use PSM and stratified analysis to exclude the influence of this segment of the population and study their specificity. We have added instructions and highlighted them in red in the original text.

Include a supplementary table with FFQ items and portion-size estimation method.

Thank you for your professional and constructive comments. We supplement this in the Supplementary material.

Include the references of the FFQ.

Thank you for your professional and constructive comments. We have supplemented this section and marked it in red.

Specify if were contracted dietitian to gather the information.

Thank you for your professional advice. We have supplemented it and highlighted it in red.

Include the information of the anthropometric equipment (e.g. brand, model).

Thank you for your professional advice, which we have supplemented and marked in red in the original text.

Justify sampling exclusions and PSM caliper value (0.25).

Thank you for your professional advice. According to your suggestion, we have added this part of the description in methodology and marked it in red.

Results

Include the information related to the statistical analysis to stablish the statistical signification at the bottom of the table.

Thank you for your professional advice. I have added this part to the paper and marked it in red.

Discuss overrepresentation of urban vs. rural participants (40.6% rural) and its impact on generalizability.

Thank you for your professional advice. I have added this part to the paper and marked it in red.

Provide p-values for subgroup analysis (e.g. low-protein intake group)

Thank you for your comments. P values are provided in the forest plots.

Clarify why “normal weight” subgroup showed stronger association.

Thank you for your comments. We have added this part of the discussion in the discussion and marked it in red.

Discussion

Compare findings to paediatric-specific studies (e.g. branched chain amino acids in adolescents).

Thank you for your professional and constructive comments. According to your comments, we have supplemented it and marked it in red.

Limitations

Add a dedicated subsection titled “Limitations”. Discuss residual confounding (e.g. genetic factors, mental health).

Thank you for your professional and detailed opinions. According to your requirements, we have added relevant content in the original text and marked it in red.

Address FFQ’s inability to capture real time dietary changes.

Thank you for your professional and detailed opinions. According to your requirements, we have added relevant content in the original text and marked it in red.

Best regards,

Reviewer 3 Report

Comments and Suggestions for Authors

Dear Authors

1. Here is a similar literature "Prospective associations between dietary patterns and high sensitivity C-reactive protein in European children". Please, explain the difference between yours and the previous literature

2. The introduction part was too short. The authors should rewrite or modify more sentences with references. 

3. Font size was too small in Figures 1, 2, 3, 6 and 7. Please, enlarge the size of the words. 

4. The discussion part was too short. Please, discuss your conclusion with more references. 

5. The title of this manuscript was ambiguous. Thus, clarify or rewrite the title based on your results. 

Author Response

Dear Authors

  1. Here is a similar literature "Prospective associations between dietary patterns and high sensitivity C-reactive protein in European children". Please, explain the difference between yours and the previous literature

Answer: Thank you for your professional and detailed comments. What is different from this literature is that our study focused on the relationship between dietary amino acids and hs-CRP. However, much of this literature has focused on children's dietary patterns. In this study, we studied the secondary structure of the protein and further subdivided the protein to explore the amino acid composition of the hs-CRP abnormality. Thank you for your comments and concern.

  1. The introduction part was too short. The authors should rewrite or modify more sentences with references.

Answer: Thank you for your professional and detailed comments. According to your comments, we have added and modified the introduction. At the same time, we refer to the previous pages of this magazine to ensure that we meet the requirements of the magazine. Thank you again for your professional and constructive comments.

  1. Font size was too small in Figures 1, 2, 3, 6 and 7. Please, enlarge the size of the words.

Answer: Thank you for your professional and detailed comments. According to your comments, we have revised it. Thank you again for your comments.

  1. The discussion part was too short. Please, discuss your conclusion with more references.

Answer: Thank you for your professional and detailed comments. According to your comments, we have revised it and marked it in red.

  1. The title of this manuscript was ambiguous. Thus, clarify or rewrite the title based on your results.

Answer: Thank you for your professional and detailed comments. According to your comments, we have revised it and marked it in red.

Round 2

Reviewer 1 Report

Comments and Suggestions for Authors

As Authors have responded sincerely and appropriately to my comments, I am inclined to accept it.

Author Response

Thank you for your professional and constructive comments. Your comments have greatly improved the quality of our paper. Thank you again for your guidance and affirmation.

Reviewer 2 Report

Comments and Suggestions for Authors

Dear Authors,

Thank you for the improvements made so far.

I have one last request: please include, at the bottom of all tables, information regarding the statistical analysis used. For example:

1 Student’s t-test for independent samples (p < 0.05); 2 Student’s t-test for paired samples (p < 0.05); 3 Pearson Chi-square (p < 0.05); 4 McNemar’s test (p < 0.05).

The numbers (1, 2, 3…) should be included only if more than one statistical test is applied within the same table. If a single test is used, there is no need to number it.

https://www.mdpi.com/2227-9032/12/9/942

Best regards,

Author Response

Thank you for your professional and detailed advice, your guidance is very useful. We have modified it according to your request. Thank you again for your guidance.